# Comprehensive Evaluation Method of Historical Timber Structural Building Taking Fujiu Zhou House as an Example

**Peixuan Wang** [1,2,*] , **Shengcai Li** [1] , **Nicola Macchioni** [3] , **Sabrina Palanti** [3] and **Gabriele Milani** [2]

1   College of Architectural Science and Engineering, Yangzhou University, Huayang Avenue 198, Yangzhou 225127, China; lisc@yzu.edu.cn
2   Department of Architecture, Built Environment and Construction Engineering (ABC), Politecnico di Milano, Piazza Leonardo da Vinci 32, 20133 Milan, Italy; gabriele.milani@polimi.it
3   CNR-IBE, Institute of Bioeconomy, Via Madonna del Piano 10 Sesto Fiorentino, 50019 Florence, Italy; nicola.macchioni@ibe.cnr.it (N.M.); sabrina.palanti@ibe.cnr.it (S.P.)
*   Correspondence: peixuan.wang@polimi.it; Tel.: +39-3488803610

**Abstract:** Physical and mechanical properties of timber components are the basis of developing the technical measures for the conservation and restoration of historical timber structural buildings. By means of integrating on-site investigation (such as a visual survey, moisture content test, micro-drilling resistance test, and material samples collection of historical timber components) and laboratory tests, this study proposed a series of methodologies for comprehensively evaluating the physical and mechanical properties of timber. This method can be quickly mastered by various non-professionals and can help the cross-learning of various disciplines engaged in the research of architectural heritage protection. As a trial, the methodologies were applied to survey and assess a typical historical Chinese timber structural building named the Fujiu Zhou house (the house is located in No. 19, Qinglian lane, Yangzhou city, Jiangsu province, China). The paper studies the 224 components of the main structure of the building, including 128 columns and 96 beams. With the help of the components' defects and damage status, GB/T13942.2-1992 and the National Lumber Grades Authority (NLGA), the grade of timber components was distinguished. The modulus of elasticity (*MOE*), modulus of rupture (*MOR*), and other related material properties parameters of timber components were also obtained. The trial results verify that the proposed methodologies are reasonable, and they can be helpful for the conservation of a historical timber structural building.

**Keywords:** historical timber structural building; on-site investigation; defects and decay; physical and mechanical properties; comprehensive evaluation

## 1. Introduction

Timber structural building is an important part of traditional Chinese architecture. It was used not only in monumental buildings such as palaces, temples and pagodas, but also in some less known residential buildings. Through their aesthetic values, structural characteristics, artistic expression and other aspects, the precious influence of timber structure on Chinese history can be explored easily. Due to the characteristics of timber, which is biodegradable, a lot of traditional timber structural buildings are in urgent need of protection.

A traditional Chinese timber structure is mainly built by frames, then purlins and longitudinal beams connect these frames to complete the whole structure of a building (Figure 1). The most common types of frames in the Jiangnan area of China are post and lintel construction frame and column and tie construction frame. Post and lintel construction frame (Figure 2a) places beams on columns and then places short columns on beams. The weight of the roof is transferred to the foundation through rafters, purlins, beams and columns. This kind of construction frame has large span beams, and it provides more space to the building. It was generally applied to historical palaces and temples, and

most of these timber buildings have been protected well because of the importance of their status and culture. Column and tie construction frame (Figure 2b) uses square beams to string up the columns, its purlins are placed directly on top of the columns. The weight of the roof is transferred to the columns through purlins directly. This kind of construction frame has smaller inner spaces, lower requirement of timber material, and lower cost. Column and tie construction frame has a high degree of acceptance among people, it is usually used in traditional residential buildings, and some of these buildings have not yet been adequately protected and valued.

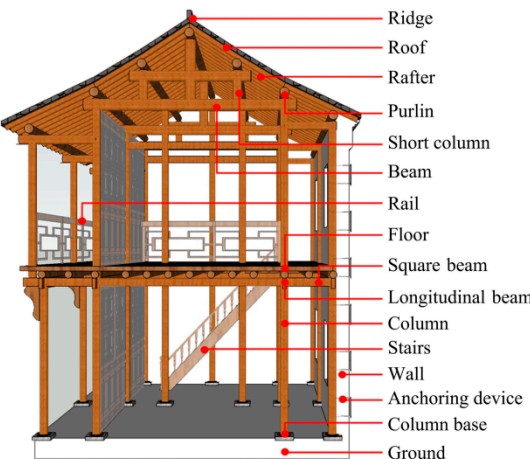

**Figure 1.** Traditional Chinese timber structural building.

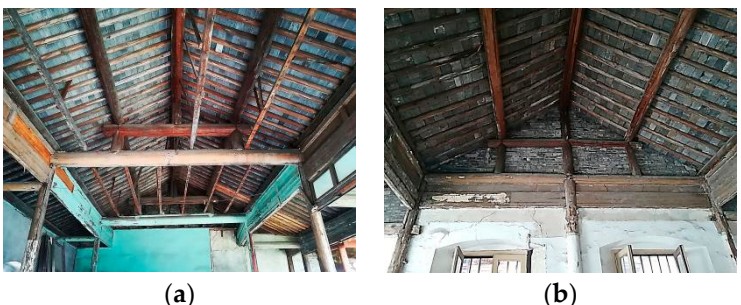

**(a)**            **(b)**

**Figure 2.** Structure types; (**a**) post and lintel construction frame; (**b**) column and tie construction frame.

The analysis of the protection of historical timber structure heritage is based on the whole structure and then develops to single components. The research method is also gradually developed from the simplest appearance analysis to on-site testing, precision instrument testing, and various digital technology studies. As for the whole timber structural building, in addition to the damage characteristics of the material itself, different construction methods and joint connection methods also have a profound impact on the durability of the architectural heritage. For the timber components, historical timber buildings are sometimes constructed by a variety of different timber species, and components contain natural defects such as knots, cracks, and grain deviations. During the usage period, components and joints could also accumulate damages. These defects and damages influence timber structural performance [1,2]. Therefore, identifying the species, defects, and damage of timber components allows a comprehensive evaluation of timber structural performance. Then, evaluating physical properties and mechanical residual performances of such historical timber components is the key to defining restoration methodology, protection means and maintenance operations to extend the durability of timber structures.

The research on the durability of a whole timber structural building has been the focus of scholars' long-term attention. Based on the current research status at home and abroad,

combined with the application of historical timber structures, Dong et al. compared several commonly used loss identification methods, and analyzed the main problems of damage identification methods in a whole structure [3]. Li et al. explored a method of estimating the remaining life of a structure through the study of the load-bearing capacity and durability of the timber structure, which is of great significance to the repair and maintenance of a historical building [4]. Ni et al. took a traditional historical timber structural building as an example, started from the overall characteristics and current situation of timber structure, and studied the damage reasons and corresponding repair methods of the timber structural buildings [5]. Wang conducted analysis and research on the historical timber structural building systems in China and Germany, and deeply analyzed the damage characteristics of timber structural buildings at the material and structural levels. Comparing the experience of the protection technology between the two countries, the researcher concluded that the protection technology of Germany is relatively advanced, which is worth learning from, and making use of and innovating [6].

In the aspects of detection technology and evaluation methods of timber components, scholars have performed a lot of research and practice. Frank Rinn invented the micro-drilling resistance test method which is used to detect timber and its internal structure [7,8]. Nowak et al. presented a survey of the state-of-the-art application of drilling the resistance method as an almost non-destructive (semi-destructive) diagnostic technique for testing timber structures, with examples of their application. They also researched using a mobile micro-drilling resistance meter to investigate the technical state of buildings, including those of high historical value [9]. Zhang et al. explained the basic principle of micro-drilling resistance methods, provided the resistance curve form of different types of timber structural defects, and determined the evaluation methods of them [10]. Liao et al. studied how to use a stress-wave tomography meter and a micro-drilling resistance meter to test the internal defects of cylindrical timber components [11,12]. Dai et al. studied the beams and columns of historical timber structures through a stress-wave tomography test and micro-drilling resistance test. At the same time, they compared the results from the two tools, then provided new ideas for the repair of historical timber structures [13,14]. Cruz et al. described a survey methodology supported by non-destructive testing to analyze the timber components which are not directly visible. The methodology is aimed at evaluating the original characteristics of each component and the possible modifications occurred during the service life of the structure such as biological decays and structural damages [15]. The Italian standard UNI 11119:2004 cultural heritage-wooden artifacts-load bearing structures-on site survey for the diagnosis of timber components was the first official document describing a survey methodology for historical timber structures [16]. The visual and non-destructive testing methods described in the standard can provide reliable data for the diagnosis of the components' situation. Clearly, the methodology is applicable to the timber species most used in Italy and to the local timber structure typologies. Macchioni et al. applied the described visual and instrumented methodology on the timber roof structure of a church outside Europe [17]. Chinese standard GB50165-92 technical specification for maintenance and reinforcement of historical buildings provides guidelines to be followed to maintain and restore historical timber structural buildings [18]. Finally, the European standard EN 17121:2019 was published in October 2019, and it describes the assessment methodology of historical timber structures [19].

The above achievements provide technical support for the exploration and performance evaluation of historical timber structures from different levels. However, it is found that the existing research methods mostly rely on more professional material testing and structural knowledge, and the research cycle is long, which cannot be mastered by non-professional researchers. The protection and research of ancient timber structural buildings is a very broad subject. It emphasizes the interdisciplinary research of architecture, structure, history, material science and other disciplines. Therefore, it is very important to summarize a set of technical methods for estimating the physical and mechanical properties of timber components that can be quickly learned and used by various non-professionals.

This study used this as a starting point, summarized the methods of estimating the physical and mechanical properties parameters of historical timber structures based on the results of on-site investigation and comprehensive evaluation. The specific methodology is shown in Figure 3.

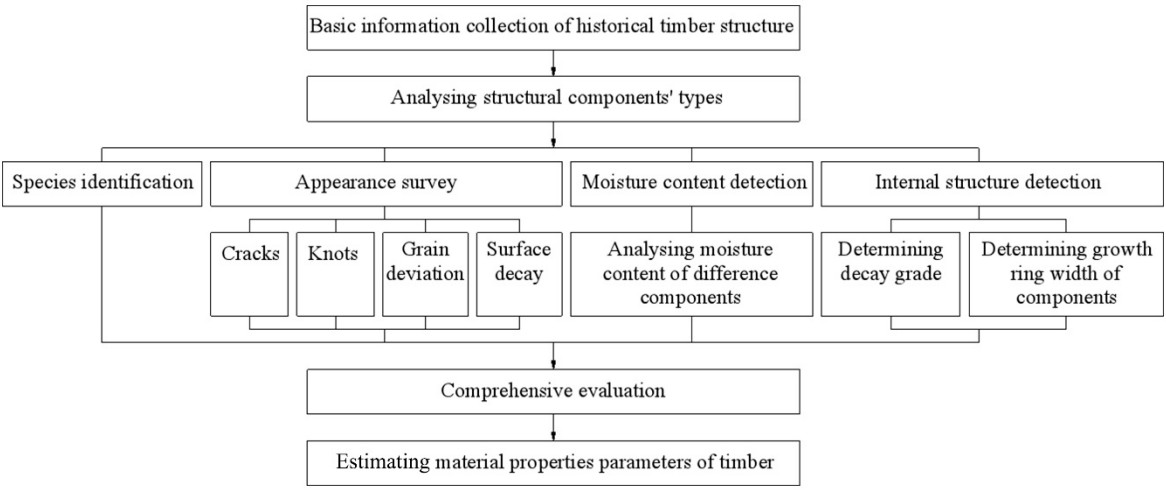

**Figure 3.** Flowchart of comprehensive evaluation of historical timber structural building.

## 2. Methodologies

### 2.1. Species Identification and Appearance Survey

The methodology for the identification of timber species is according to the Italian standard UNI 11118:2004 cultural heritage-timber artifacts-criteria for the identification of timber species [20]. It was carried out by two steps. Firstly, the macro characteristics of the structure timber components were analyzed to determine whether they are softwood or hardwood. Secondly, a sample was taken from components with similar macroscopic characteristics to identify the species through anatomical analysis. The sample was cut to slices manually after it was boiled in water, then the slices along the three major directions (cross, radial and tangential sections) were observed by transmission optical microscopy. To identify the timber, the observed anatomical features should be compared with the information from the major timber atlases (in the specific case studied in the following, reference [21] mostly useful).

Visual survey is the first step in the detailed survey of timber structures. Cracks, original defects and decay are three basic information that can be identified and measured by the naked eye during visual survey. Because of its own material properties, timber almost always contains cracks and knots. Shrinkage cracks are generally divided into many types, and they are unavoidable when components contain the pith. However, they do not significantly reduce the mechanical properties of the timber components, especially in columns which are compressed vertically. Knots and grain deviations are a natural mechanical defect in timber, and are major factors in assessing the grade of timber. In the appearance survey, the perimeter of the component, the diameter of the largest knot, the length, width and depth of the crack, the angle of the grain deviation and the position of the surface decay should be recorded.

## 2.2. Moisture Content and Internal Structure Detection

According to European standard EN 335, researchers can perform the evaluation of the natural durability of timber. The measurement of the moisture content of timber components was effectuated for evaluating the use classes of timber components and if the natural durability was sufficient [22]. In the definition of use class, the disposition of some components of historical buildings that are connected to the wall, put into the ground or exposed to the atmospheric conditions, influenced the moisture content of timber components and consequently their conservation. First, it is recommended to use a wood moisture meter that is easy to carry and has high sensitivity. As shown in Figure 4c, this kind of detector mainly uses the sensor on the front of the instrument to detect moisture content. The location of the measuring points should be selected at the base, half-length, and the top of columns (Figure 4a), as well as the two ends and half-span of the beams (Figure 4b), Measurements were taken from different directions of the same section, then the average values were calculated. In addition, traditional timber structures commonly use round wood as components, and there is a difference in the contact process between the detector and the circular surface. Therefore, the same section should be measured from different directions and averaged to reduce the impact of the difference. Generally speaking, it is recommended to perform detection in three directions on the same cross-section plane, each direction being 120° from each other (Figure 4d).

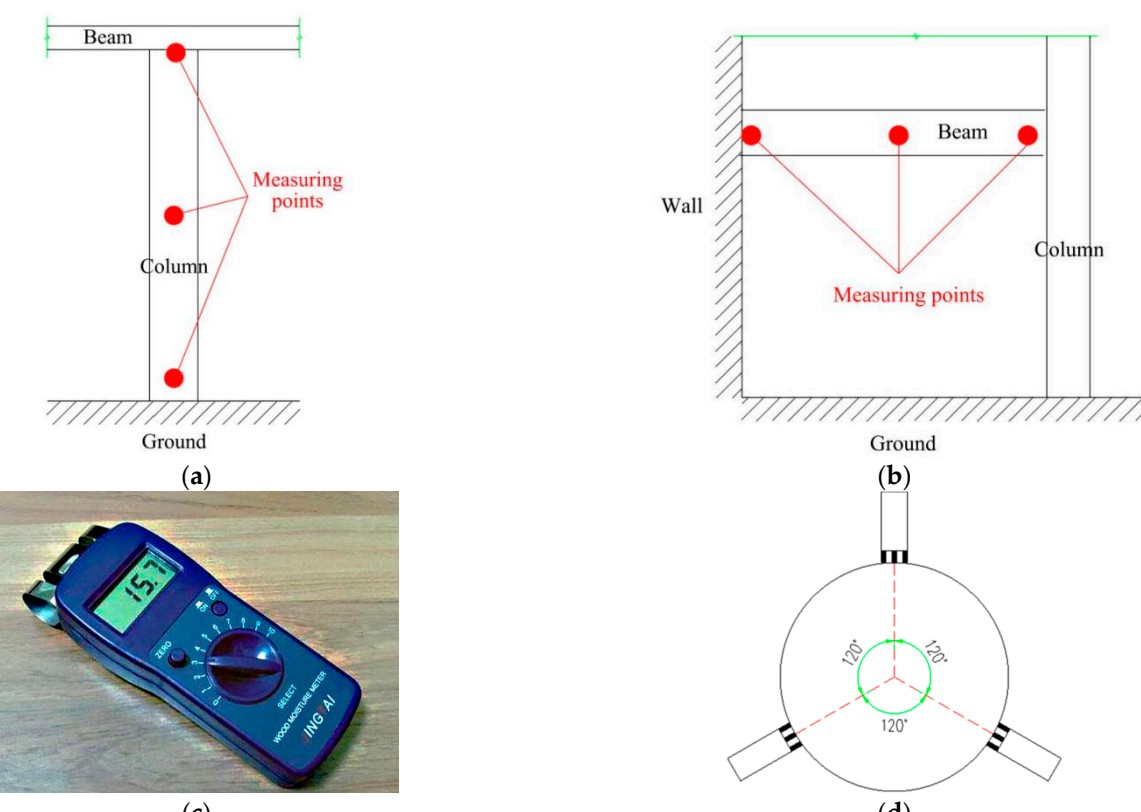

**Figure 4.** Moisture content detection method; (**a**) measuring points of column; (**b**) measuring points of beam; (**c**) wood moisture meter JK W10; (**d**) measuring points in the same cross-section plan of one component.

The micro-drill resistance instrument (Figure 5) can help us to complete the investigation of the internal condition of the component under the precondition of protecting the integrity of the timber component to the greatest extent. The working principle of it is to drive the drill needle into the interior of the timber at a constant rate through the motor drive, and the corresponding resistance is generated during the drilling process. According

to the micro-drilling resistance map, the internal damage of the component, the growth ring width (GRW) and other structural conditions can be quickly evaluated and judged.

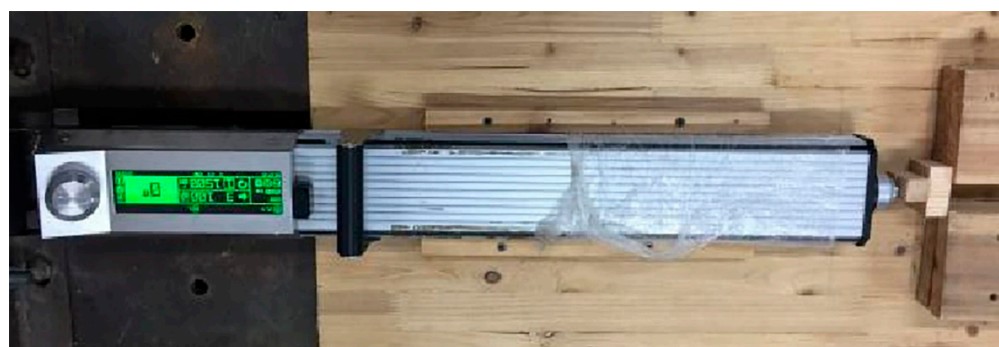

**Figure 5.** Micro-drilling resistance instrument IML PD400.

When doing a micro-drilling resistance test of the columns, the columns should be divided into two categories. For the same inspecting section of the columns that are completely exposed to the outside, select three different drilling directions separated by an angle of 120° from each other (Figure 6a). For the columns that are half exposed, one or two directions are selected according to the specific situation (Figure 6b). For the same column, take the 5 cm distance from the ground as the first testing section. If damage is detected, continue detecting by moving the same scheme upward by 5 cm or 10 cm until no damage is detected. The beam detection method refers to the column detection method. The condition of the timber structure at the beam-column joints needs attention, and the inspection is usually carried out at the angle of 45° between the beam-column joint (Figure 6c). Currently the resistographic analysis is used to investigate the internal decay of timber structures, based on the evaluation of the resistance profile during the needle perforations. The registration of resistance profiles permits also to quote the decay, based on the percent of soundness or decay in it.

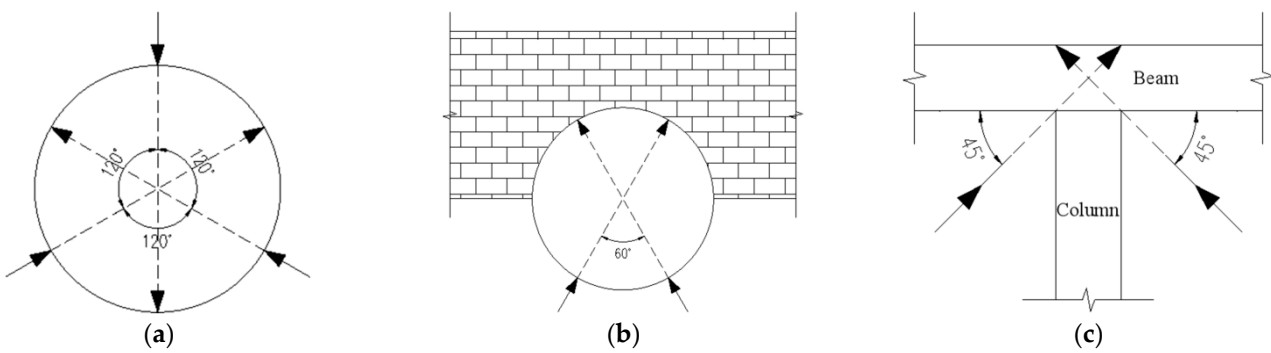

**Figure 6.** Internal damage detection method; (**a**) fully exposed column; (**b**) half exposed column; (**c**) beam-column joint.

After the end of the test, the internal damage of the timber components can be determined by the instrument display spectrum. The resistance maps of the timber components without damage are more stable and uniform, the frequency is faster, the cycle is stable, and the peak-wave trough phenomenon continuously appears. At the crest of the curve, the density of the timber is higher, and it is latewood. At the trough of the curve, the timber density is smaller, and it is earlywood (Figure 7a). When there are decayed areas in the timber, the mechanical properties change, the strength and hardness decrease to varying degrees, and the resistance to the needle also decreases. As a result, the resistance map of the area appears to fluctuate and it declines more obviously with a slower slope area in the process (Figure 7b). If there are hollows or cracks in the timber structure, the needle suddenly receives no resistance in this area, the curve drops steeply, and the map appears

as a cliff (Figure 7c). The resistance curve of the micro-drill suddenly rises from the normal level when there is a knot in the timber, and the overall level is equivalent to several times the normal level. The knots affect the following resistance maps, resulting in the rise of the map, and even the curve of the decayed part may show a normal level (Figure 7d).

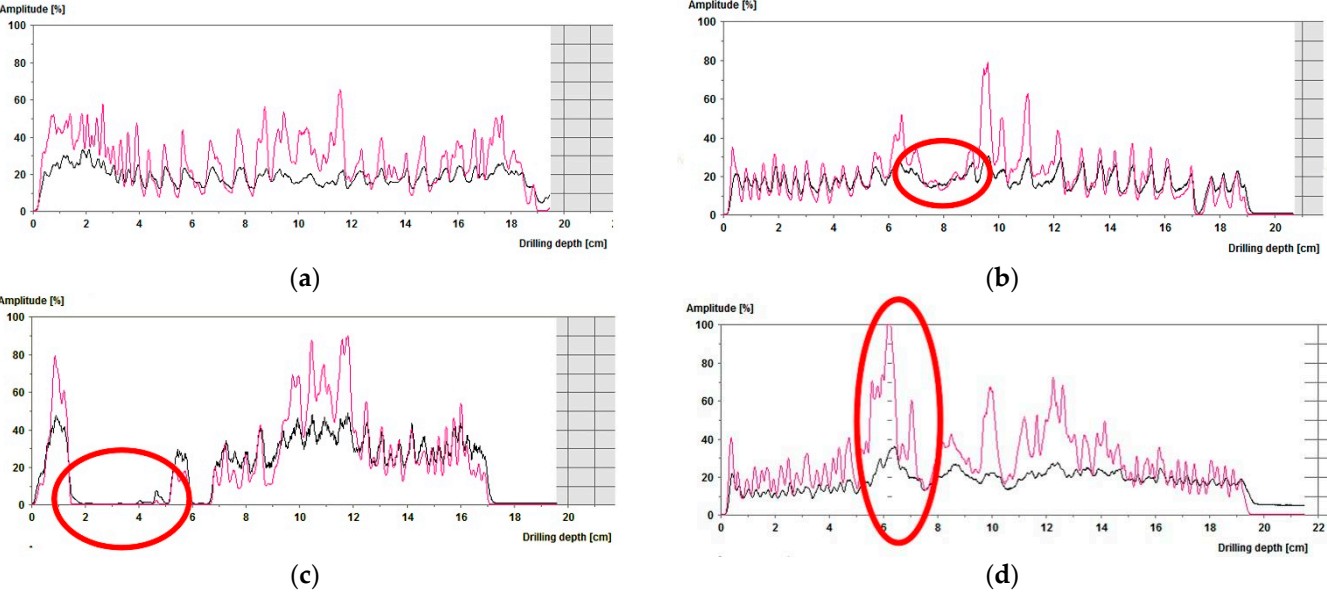

**Figure 7.** Method for judging micro-drill resistance maps; (**a**) map with good internal structure; (**b**) map with internal decay; (**c**) map with internal hollows or cracks; (**d**) map with internal knots.

### 2.3. Comprehensive Evaluation

Species identification is the first step in evaluating the comprehensive condition of timber components. The results are helpful to the analysis of the physical and mechanical properties of timber. The properties of different kinds of timber vary greatly, and the relevant research results are generally based on a specific kind of timber as the research object. The knowledge of wood species and of its biological resistance is a first answer to the status of conservation of a timber structure. The property, the constructive technology and the project details, the local climatic (macroclimate and microclimate) conditions, and the maintenance during the time are all the main factors influencing the state of conservation of a timber structure.

Cracks, knots, grain deviations and other defects determine the structural level of timber components. The shrinkage cracks caused by the change in moisture content has little effect on the working performance of components.

Cracks caused by structural forces have a greater impact on the shear and flexural performance of the structure.

The existing research results show that the moisture content of timber is highly correlated with the degree of fungal and insect decay [22–24]. The moisture content of different components and different parts of the same component often determines the position and degree of its possible decay. In fact, the fungal spores are ubiquitous. They can develop, colonize, and finally decay when the moisture content of the timber is between 20 and 30%. Further, insects such as subterranean termites can develop only if a certain moisture content of timber is present. More generally, all the biological agents, even the dry timber insects, prefer developing in a moderate moisture condition of timber.

The micro-drilling resistance map can accurately determine the internal structure of timber components. By micro-drilling a resistance test, the internal decay area of components can be judged comprehensively, and the decay degree of them can be graded according to GB/T13942.2-1992 [1]. The standard divides timber into five grades according

to the internal depth of decay. They are grade I (good timber) to grade V (totally decayed timer).

### 2.4. Estimating Material Properties Parameters of Timber

In terms of material properties parameters, there is a big difference between individual timber components. To accurately determine the physical and mechanical properties of the timber, it is necessary to conduct experimental tests one by one. In accordance with the requirements of historical architectural heritage protection, the research process should try not to cause major damage and impact the building structure. Considering this factor, it is not possible to test and analyze a single component, let alone infer the mechanical properties of the overall timber structure through the compressive and tensile tests of an existing building timber component.

Relevant research results show that there is a certain correlation between the GRW, appearance grade and the bending resistance of timber [25–29]. This study hopes to judge the internal structure of the timber through the micro-drilling resistance map, and then establish the correlation between the cross section of the component and the mechanical parameters. However, it is worth noting that the results of this estimation are somewhat different to a certain extent, aiming to allow more non-structural professionals to quickly and easily grasp, and building repair and protection play a supporting role.

The GRW of the corresponding parts of the timber component were calculated by measuring the peak-valley spacing of its micro-drilling resistance map. According to the National Lumber Grades Authority (NLGA) [2], the angle of grain deviation, and the diameter of the maximum knot, the components can be divided into four different NLGA grades: SS, No.1, No.2, and No.3. Among them, the SS grade is the best timber (angle of grain deviation is less than 1/12, and the diameter of the maximum knot is less than 56.25 mm).

According to the existing literature and the research results of historical timber buildings, the selection of structural materials for traditional timber buildings in China has specific requirements, and the defects such as knots and grain deviations are strictly restricted. They are basically similar to the structural material selection standard of modern timber buildings and can reach the SS grade of the NLGA standard.

Based on relevant research results and comprehensive analysis of test data, the relationship values between GRW, modulus of elasticity (*MOE*) and the modulus of rupture (*MOR*) of historical Chinese fir components (it is the most common material in Chinese traditional timber structural building) were obtained as shown in Table 1 [25]. There are many round timbers in ancient timber constructions, and the pith often does not coincide with the geometric center of the component. It is necessary to use the micro-drilling resistance map to identify and distinguish the different GRW areas of the testing section (Figure 8), and the appearance grade and GRW dimension respectively determine the relevant parameters of each area. Then calculate the average value of the relevant material parameter of the component according to the area weight. The calculation method is shown in Formula (1) and Formula (2).

**Table 1.** Effect of GRW on *MOE* and *MOR* of Chinese fir [25].

| *n* | GRW | Mean of *MOE* (MPa) | Coefficient of Variation in *MOE* (%) | Mean of *MOR* (MPa) | Coefficient of Variation in *MOR* (%) |
|-----|-----|---------------------|----------------------------------------|---------------------|----------------------------------------|
| 1 | GRW < 4 mm | 10,670 | 14.46 | 50.0 | 23.77 |
| 2 | 4 mm ≤ GRW < 6 mm | 10,240 | 11.68 | 44.6 | 20.28 |
| 3 | 6 mm ≤ GRW < 8 mm | 8940 | 14.77 | 37.5 | 19.73 |
| 4 | GRW ≥ 8 mm | 8400 | 20.56 | 35.5 | 15.74 |

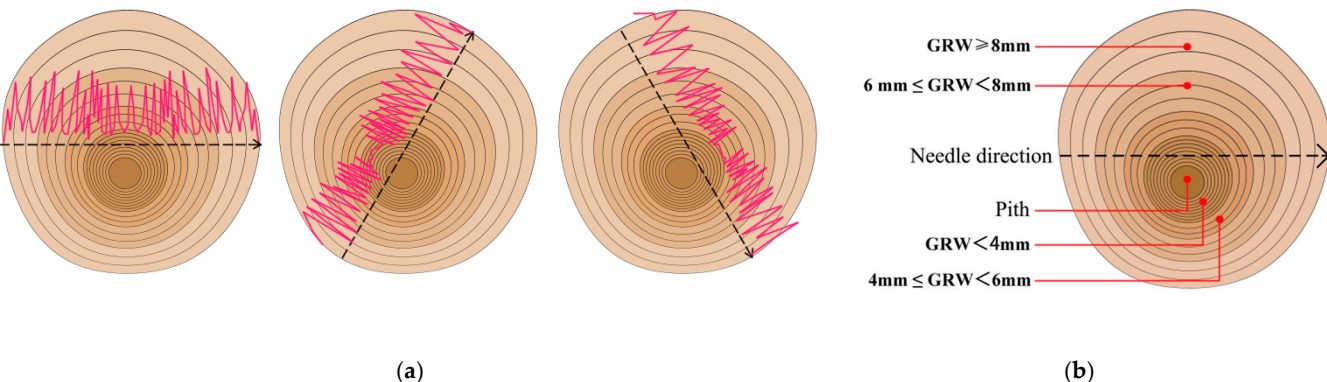

(**a**)  (**b**)

**Figure 8.** The method to divide different GRW areas in the components; (**a**) multiple direction judgment atlas; (**b**) partition area.

$$MOE = \sum_{i=1}^{4} \left( \frac{S_n}{S} \right) \times MOE_n \tag{1}$$

$$MOR = \sum_{i=1}^{4} \left( \frac{S_n}{S} \right) \times MOR_n \tag{2}$$

In the formula:
*MOE* is the modulus of elasticity;
$MOE_n$ is the modulus of elasticity corresponding to GRW, in Table 1;
*MOR* is the modulus of rupture;
$MOR_n$ is the modulus of rupture corresponding to GRW, in Table 1;
*S* is the effective section area of the component;
$S_n$ is the effective section area corresponding to GRW;
*n* is the number of different GRW, in Table 1.

It needs to be emphasized that the aim of the estimation of timber physical and material parameters is to judge the safety performance of the components. Therefore, the weakest part of the component (the most severely decayed section which is tested by micro-drilling resistance meter) is usually selected for calculation.

## 3. Results of Case Study

### 3.1. Introduction of Fujiu Zhou House

Fujiu Zhou house is a classic timber structural house built in the late Qing Dynasty (about A.D.1849–1912) which is located at No. 19, Qinglian lane, Yangzhou city, Jiangsu province, China. It has a two-floor brick-timber structure, the net depth of the building is about 73.64 m, and the width is about 50.18 m. Its total construction area is more than 3100 m². As the house was in disrepair for a long time, some parts of building have already been removed. The main structure left is a courtyard-style residence house (Figure 9). Its main timber structure are roofs, beams, columns, floors, and stairs. The house combined the two different types of construction frames, the four timber frames in the middle are post and lintel construction frames, and the others on both sides used column and tie construction frames.

Although there are a large number of timber components in the whole building, this research only considers the timber components that play a major role in structural safety, that is, the components located in the structure frames (they are all represented by the red dashed box in Figure 9a). The longitudinal beams which connect the frames in the east and west direction, timber stair, timber windows and others are not considered because they have influence of the mechanical behavior on the whole structure.

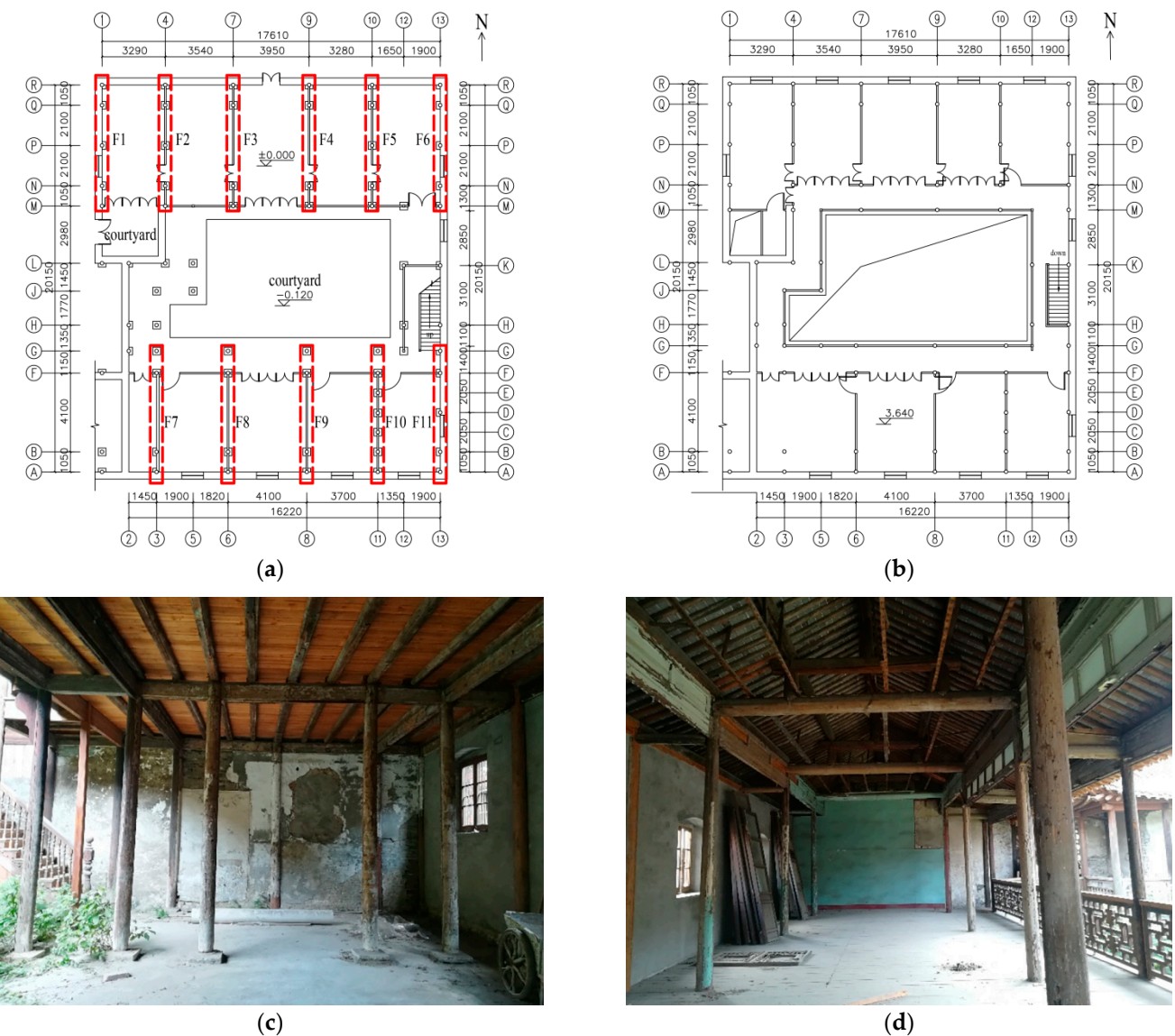

**Figure 9.** Plans of Fujiu Zhou house (the red dotted boxes show the name of frame); (**a**) the ground floor; (**b**) the first floor; (**c**) the frame structure on the ground floor; (**d**) the frame structure on the first floor.

According to statistics, there are 224 components in the considered structure. The ground floor contains 51 columns and 40 beams. The first floor contains 77 columns and 56 beams. The frames' number is shown Figure 9a, and the components' number of each frame is shown in their moisture content and decay figures.

### 3.2. Results of Application

3.2.1. Species Identification and Appearance Survey

The analysis of the anatomical features of the samples from the timber used for the Fujiu Zhou house allowed for identifying *Cunninghamia lanceolata* timber, commercially called Chinese fir. The heartwood and sapwood of Chinese fir differ significantly, the heartwood is light chestnut brown, and the sapwood is light yellowish or light grayish, growth rings are clearly visible with a gradual transition from earlywood to latewood. The main anatomical characteristic features are: axial parenchyma very abundant, dispersed similar to a star and ribbon (Figure 10a), and parenchymatic rays in tangential section are mono-seriate (rarely bi-seriate) (Figure 10b,c). Chinese fir is durable against fungal rot, and it is usually moderately resistant to termite attacks.

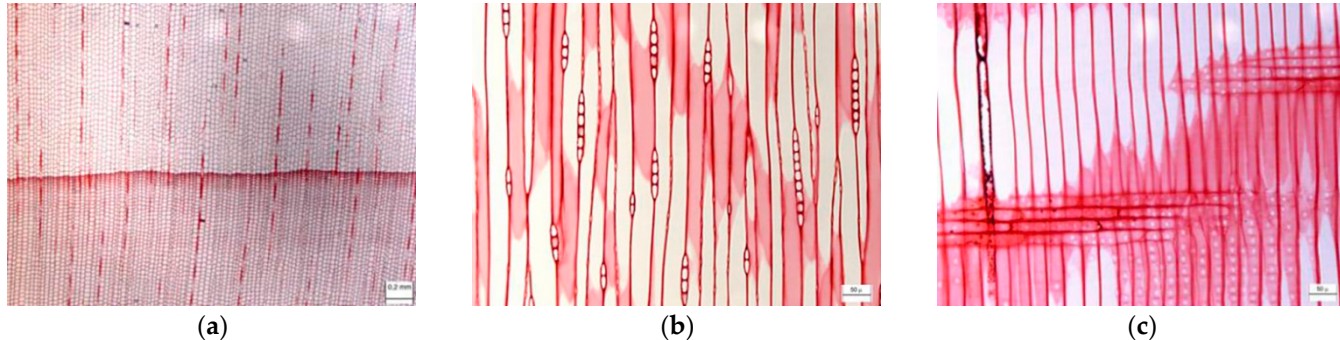

**Figure 10.** Micrograph of Chinese fir (by S. Lazzeri, CNR-IVALSA); (**a**) cross section; (**b**) tangential section; (**c**) radial section.

In the whole structure, all components have knots and cracks. The diameter of knots ranges from 9 mm to 43 mm, and about half of the cracks belong to long cracks. Their depth is up to 87 mm, and their width is up to 21 mm. The levels of components are all in SS level.

There is visible decay on some of the columns on the ground floor. The decay is basically concentrated at the bases of the columns with higher moisture content. About one-tenth of the bases of the columns have been replaced by new timber (Figure 11). The visible condition of the beam is basically intact and there is no obvious decay.

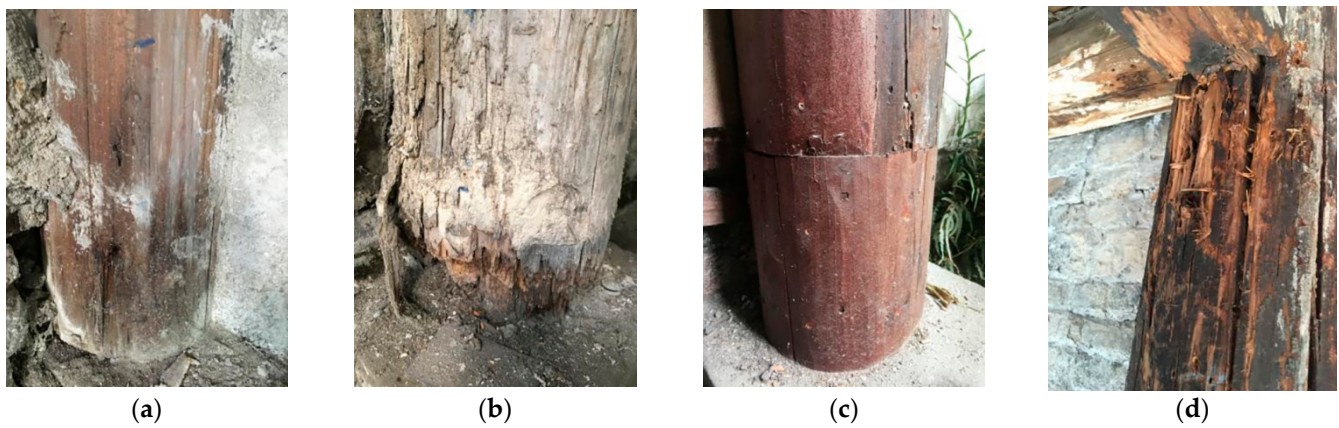

**Figure 11.** Appearance survey of Fujiu Zhou house; (**a**) column base influenced by fungi; (**b**) decay of column base; (**c**) new timber replaced the column base; (**d**) decay of the beam-column joint.

### 3.2.2. Moisture Content Test

The moisture content test of the timber components was carried out by a wood moisture meter JK W10 (Figure 4c). The test time was from 18 August 2018 to 19 August 2018, these two days were sunny summer days, and the relative humidity was 76%.

Figures 12a–22a show the moisture content of each frame with different depths of color. The darker the color, the higher the moisture content.

The moisture content of frames F1, F6 and F11 (Figures 12a, 17a and 22a) is higher than that of other frames, and there is little difference in the moisture content of different components. These three frames are located at the edge of the whole structure, and the half of their components are buried in the brick wall. The bricks commonly used in buildings in the Jiangnan area have good water absorption properties and can absorb water from the rain and air for a long time. These bricks transfer water to the frame embedded in the wall, which leads to a higher moisture content.

However, in the other frames, the moisture content difference between the components is obvious, and the moisture content of different parts of the same component is also

different. Take F5 as an example to elaborate (Figure 16a). The columns embedded in the wall (F5–C1, F5–C6) and the columns near the courtyard (F5–C5, F5–C10) have higher moisture content. For the same column, the base in contact with the ground or floor can absorb water. So, the base part shows a higher moisture content than the middle part. Similarly, the two ends of the beams that are in contact with the roof are also wetter than their middle parts. That is because the rain brings water to the beams' ends through the roof.

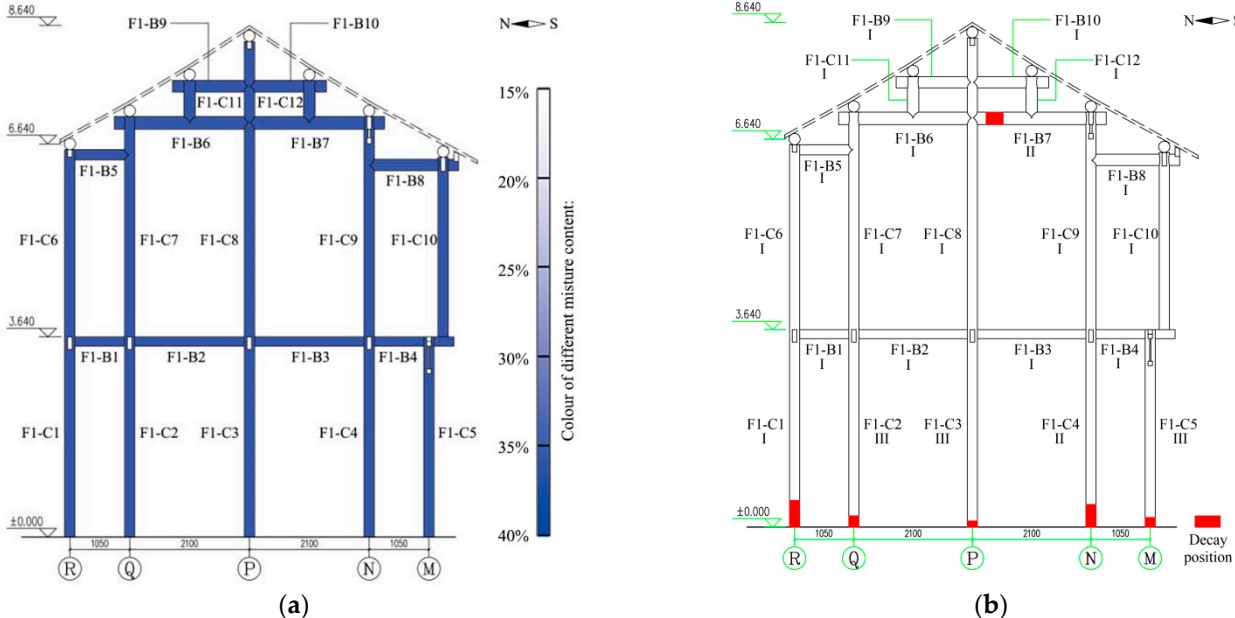

**Figure 12.** Moisture content and decay of F1 in Fujiu Zhou house; (**a**) schematic diagram of moisture content; (**b**) schematic diagram of decay position and decay grade.

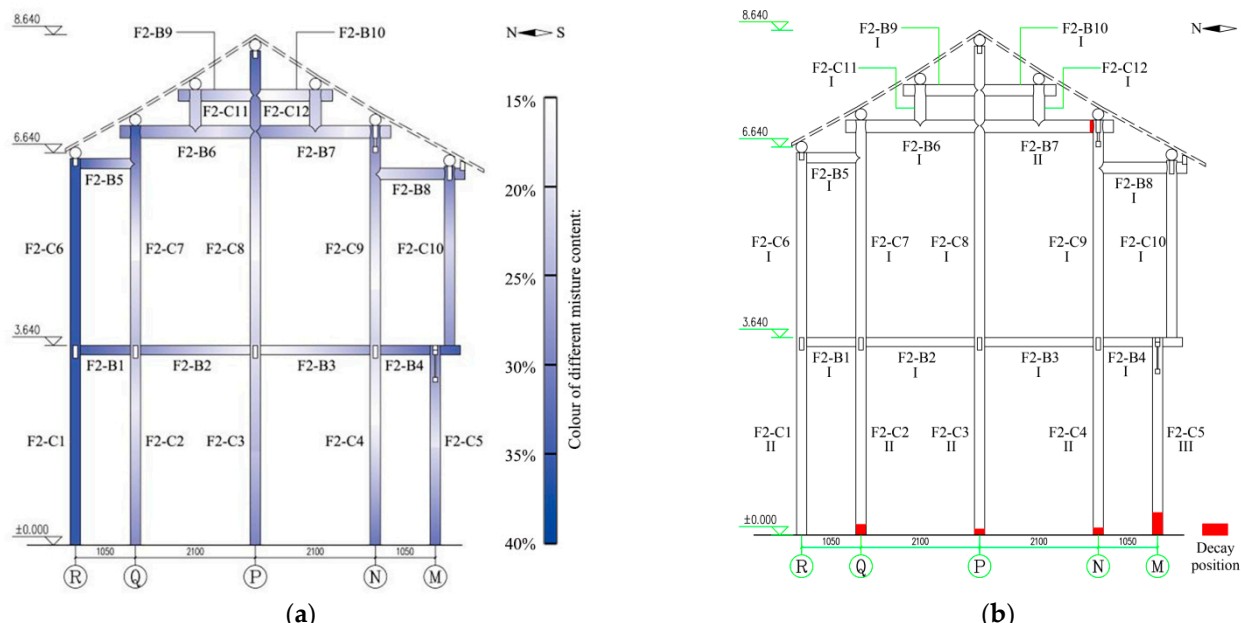

**Figure 13.** Moisture content and decay of F2 in Fujiu Zhou house; (**a**) schematic diagram of moisture content; (**b**) schematic diagram of decay position and decay grade.

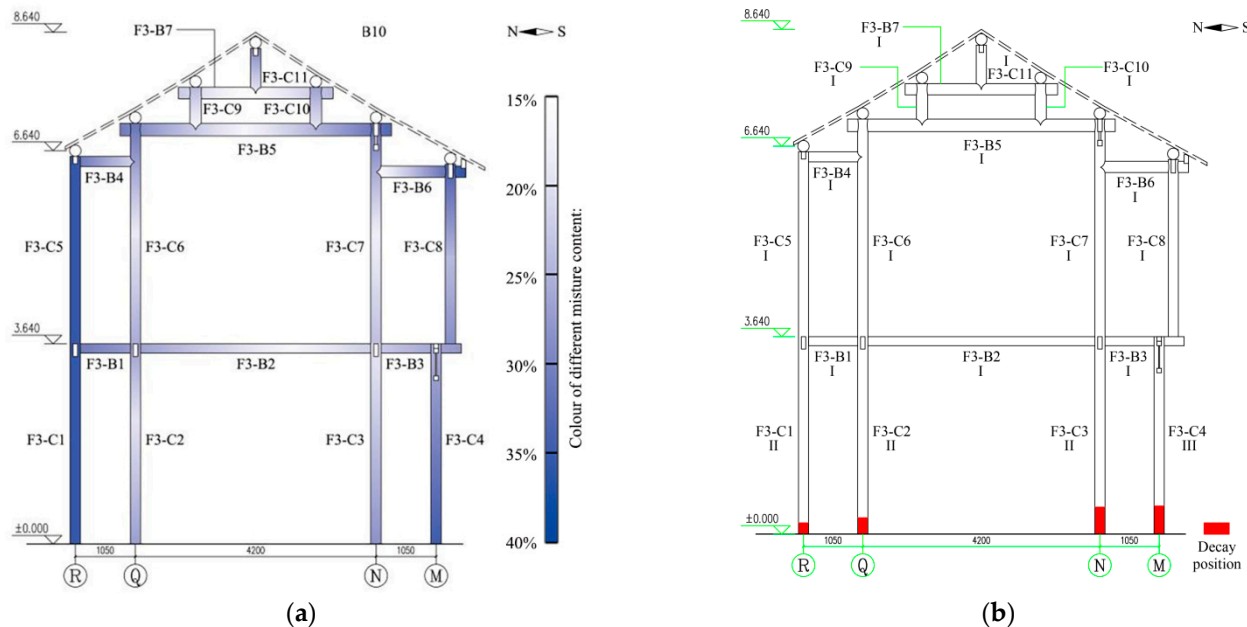

**Figure 14.** Moisture content and decay of F3 in Fujiu Zhou house; (**a**) schematic diagram of moisture content; (**b**) schematic diagram of decay position and decay grade.

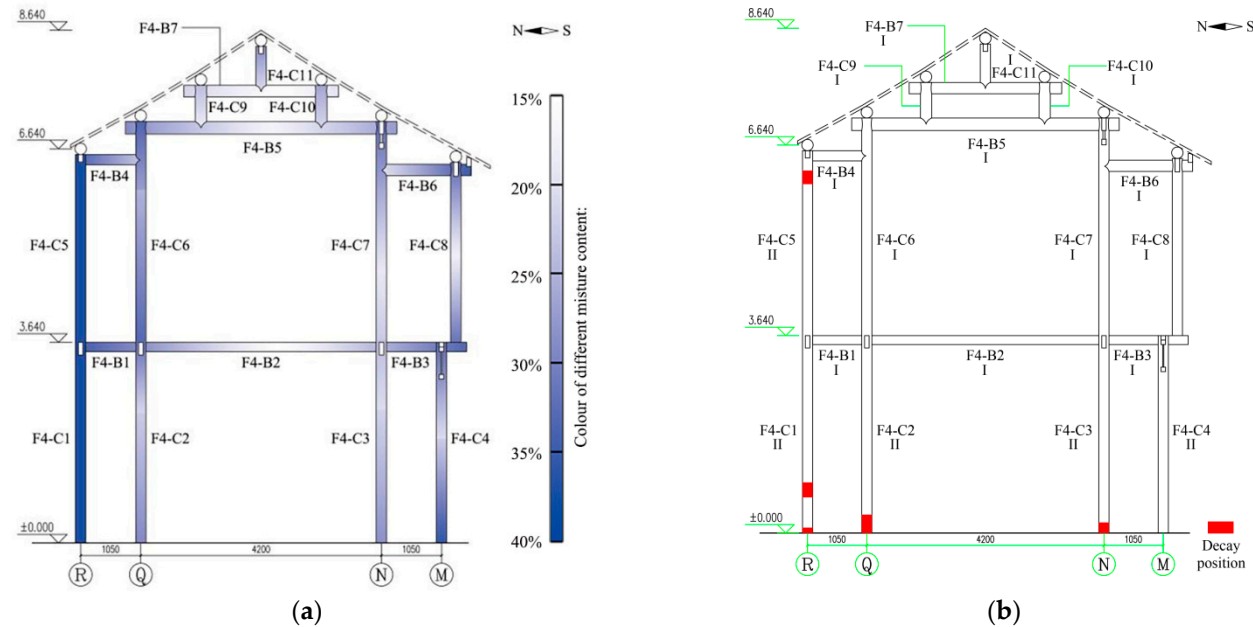

**Figure 15.** Moisture content and decay of F4 in Fujiu Zhou house; (**a**) schematic diagram of moisture content; (**b**) schematic diagram of decay position and decay grade.

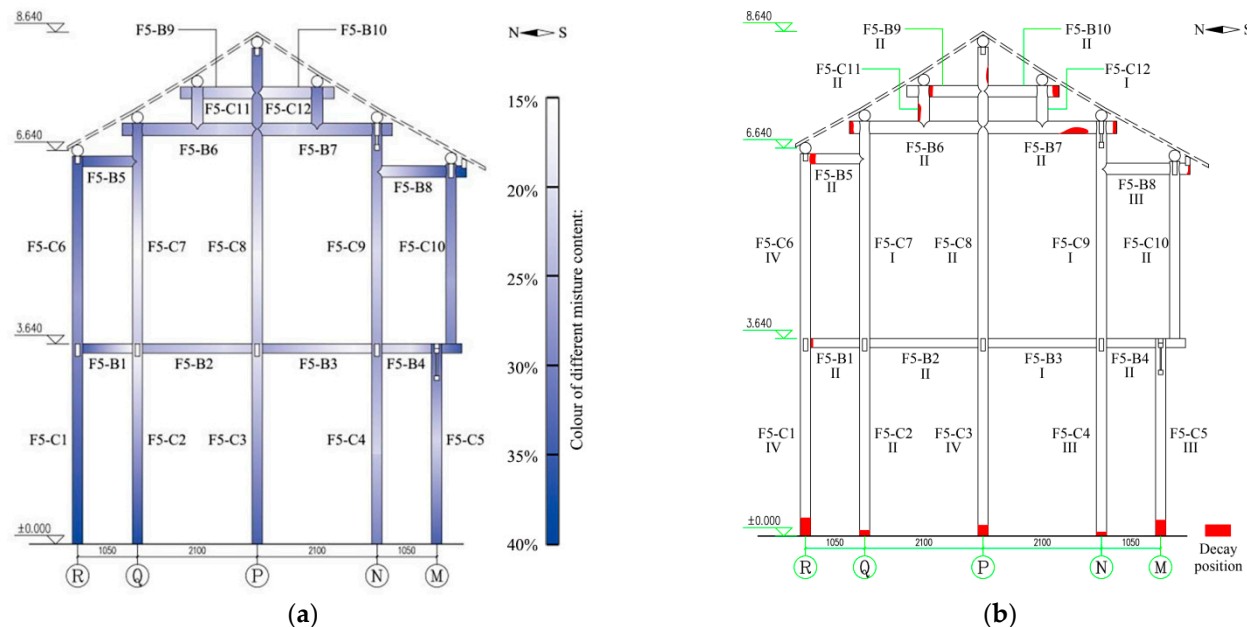

**Figure 16.** Moisture content and decay of F5 in Fujiu Zhou house; (**a**) schematic diagram of moisture content; (**b**) schematic diagram of decay position and decay grade.

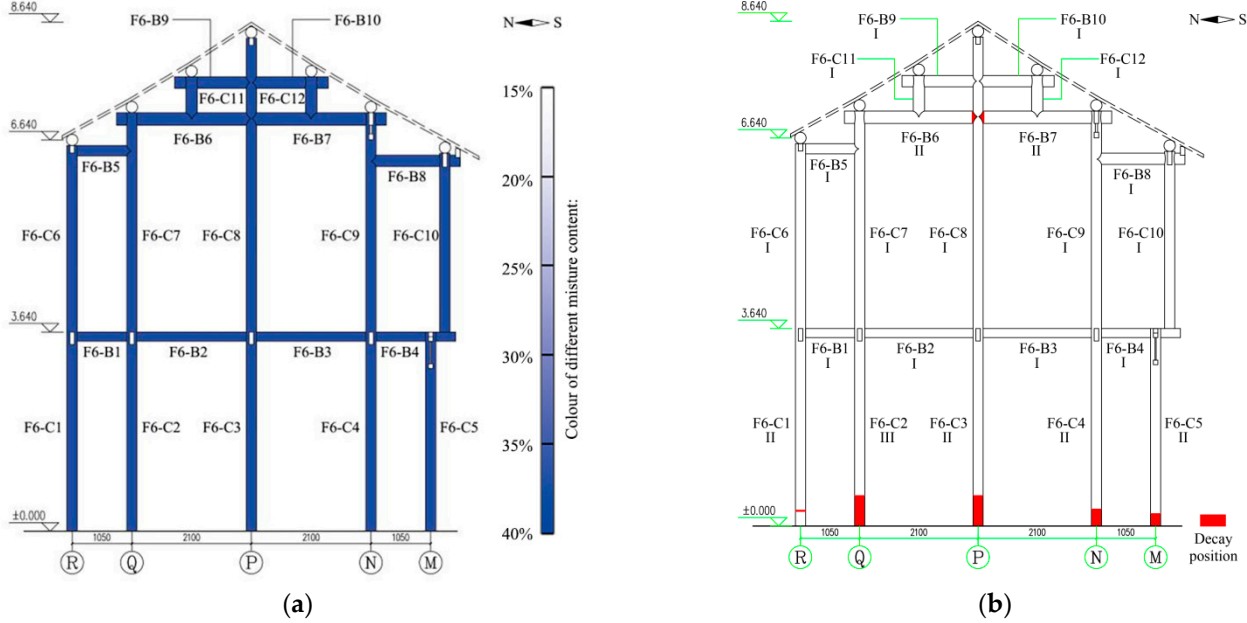

**Figure 17.** Moisture content and decay of F6 in Fujiu Zhou house; (**a**) schematic diagram of moisture content; (**b**) schematic diagram of decay position and decay grade.

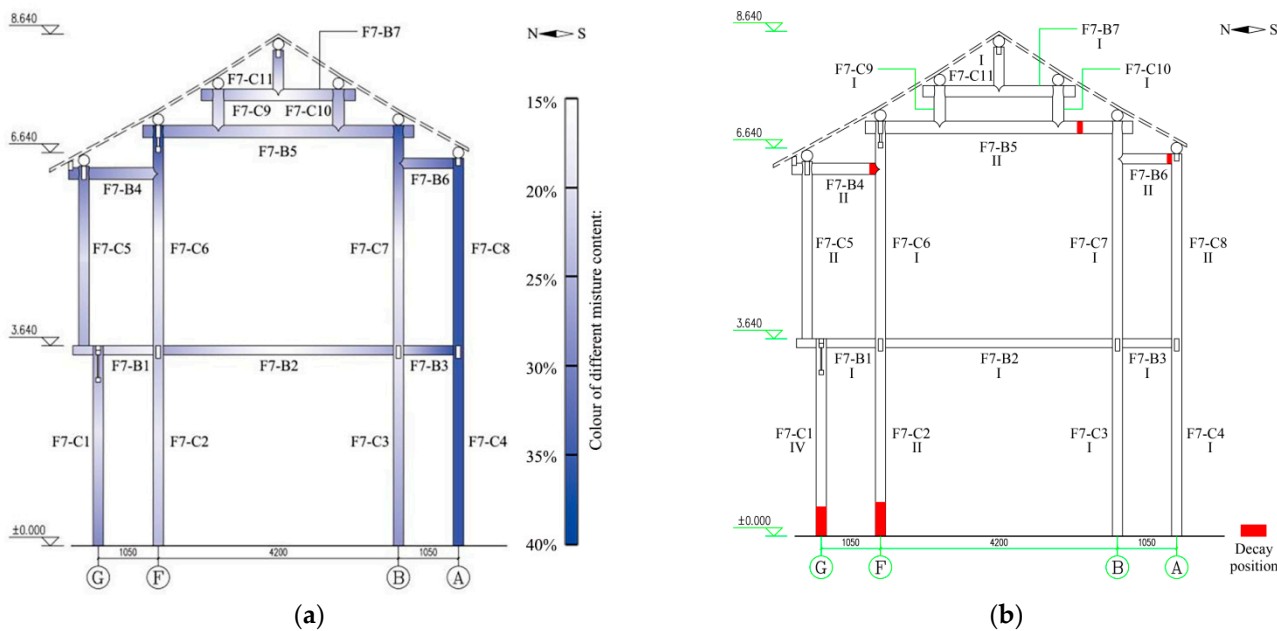

**Figure 18.** Moisture content and decay of F7 in Fujiu Zhou house; (**a**) schematic diagram of moisture content; (**b**) schematic diagram of decay position and decay grade.

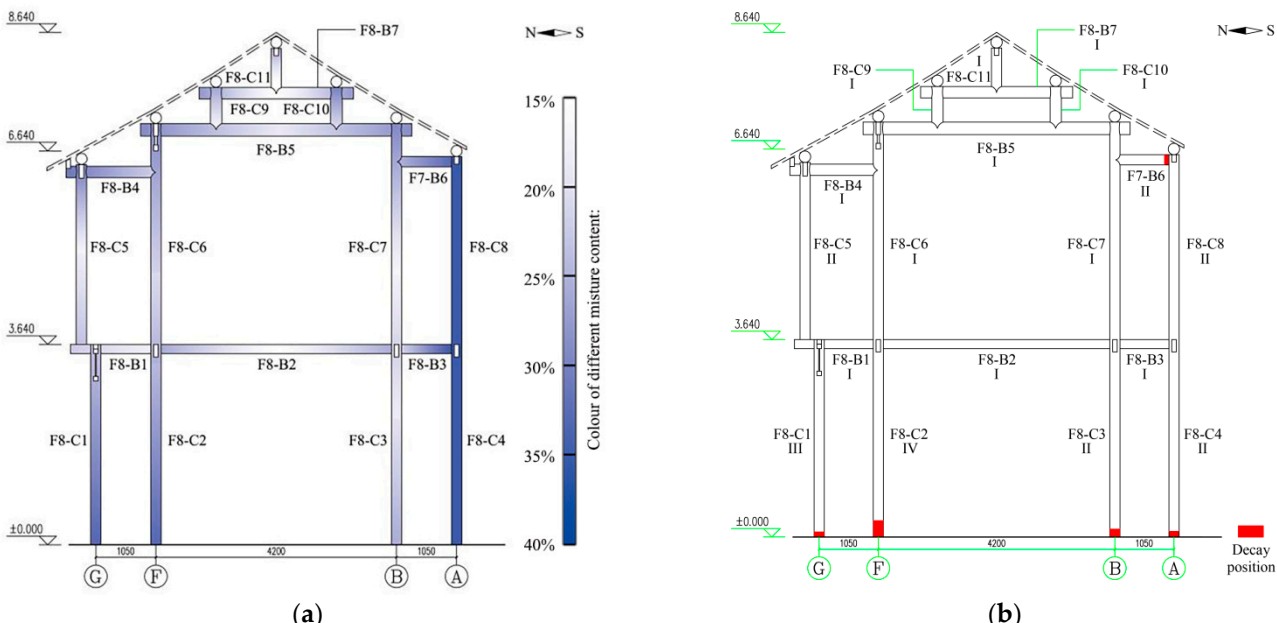

**Figure 19.** Moisture content and decay of F8 in Fujiu Zhou house; (**a**) schematic diagram of moisture content; (**b**) schematic diagram of decay position and decay grade.

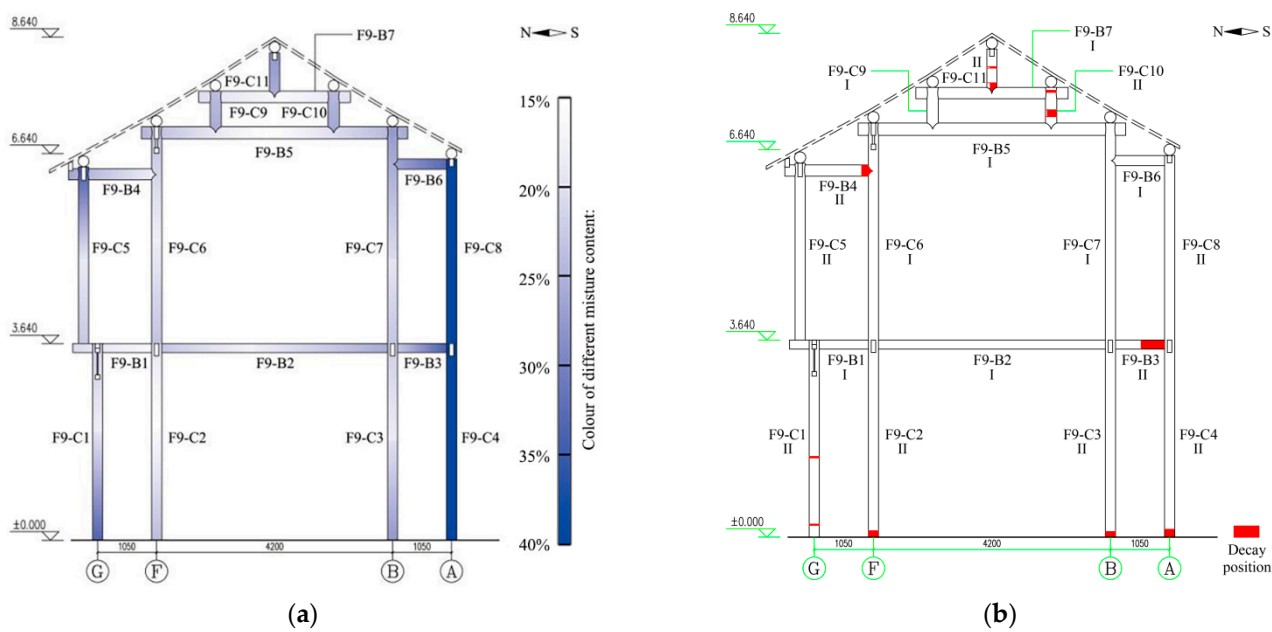

**Figure 20.** Moisture content and decay of F9 in Fujiu Zhou house; (**a**) schematic diagram of moisture content; (**b**) schematic diagram of decay position and decay grade.

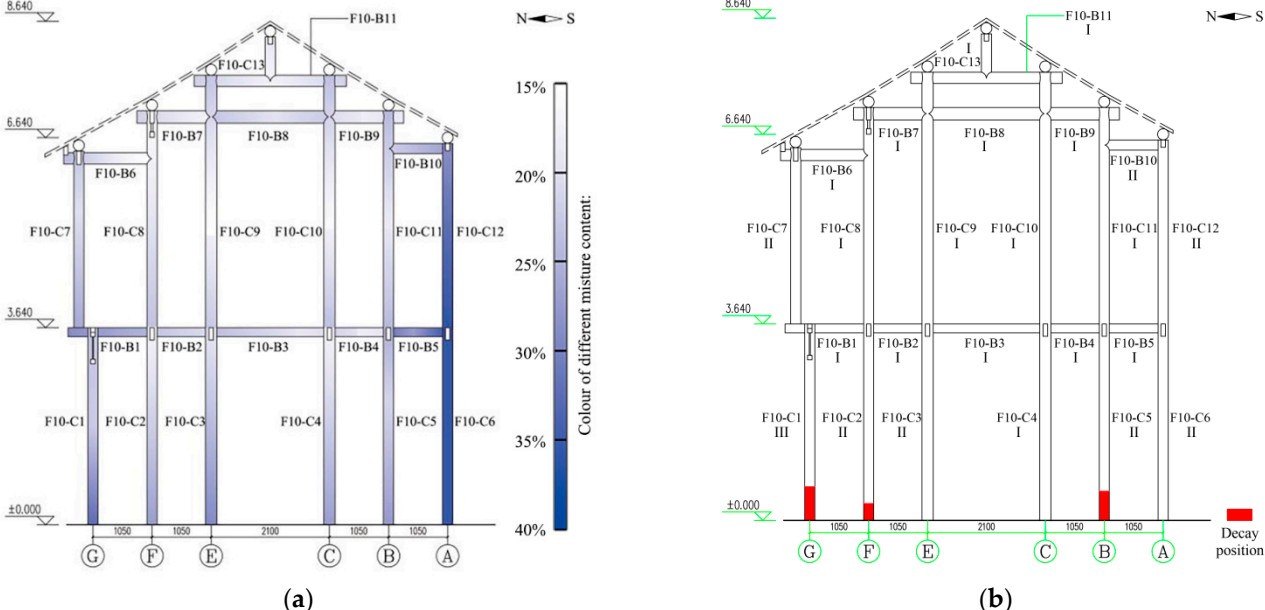

**Figure 21.** Moisture content and decay of F10 in Fujiu Zhou house; (**a**) schematic diagram of moisture content; (**b**) schematic diagram of decay position and decay grade.

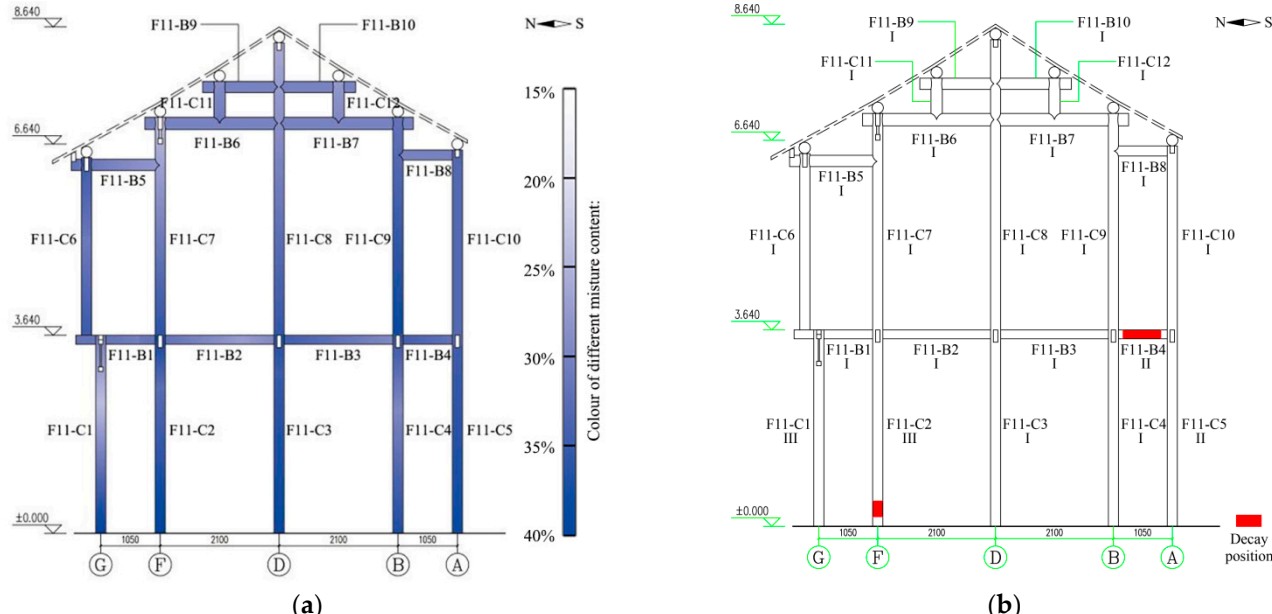

**Figure 22.** Moisture content and decay of F11 in Fujiu Zhou house; (**a**) schematic diagram of moisture content; (**b**) schematic diagram of decay position and decay grade.

In the components of all frames, the moisture content of the columns' bases is mainly concentrated between 20% and 40%, and the highest one is 39.5%. However, the moisture content in the middle of the columns is lower (between 10% and 30%), and the lowest one is 11.5%. The moisture content of the beams' two ends (north and south ends) is higher, and the highest one is 37.8%. While the middle part of the beams is drier, and the minimum moisture content is only 16%. At the same time, the moisture content of the components on the ground floor is basically higher than that of the first floor. The moisture content of the components around the courtyard, partly embedded in the wall and the beam-column joints is higher.

### 3.2.3. Defects and Decay Detection of Timber

The micro-drilling resistance instrument is IML PD400 (Figure 5). The decay position and grade of each frame are shown using red color in Figures 12b–22b. The specific detection process takes F5-C4 and F5-B9 as examples (Figure 16b).

In F5-C4, the cross section 1-1 (Figure 23a) which is 5 cm above the ground was chosen as the first micro-drilling test section. With the resistance map from three different directions, the approximate internal decay was determined (Figure 23b,c). Then, the same the same testing method was used to survey cross section 2-2 (Figure 23a) (15 cm above the ground) and cross section 3-3 (Figure 23a) (25 cm above the ground), the resistance maps depict good internal condition. Because the higher the part of the column is from the ground, the lower the moisture content, and it is less likely to decay. Therefore, the upward detection is no longer.

In F5-B9, the cross section 4-4 (Figure 24a) which is 5 cm away from the north joint, was selected as first test section. (Figure 24b,c) expresses the internal damage with the help of three different directions' resistance maps. Next, move to the south 10 cm to select the detection section. Because there is no internal decay in cross section 5-5 (Figure 24a) and cross section 6-6 (Figure 24a), the test can stop here.

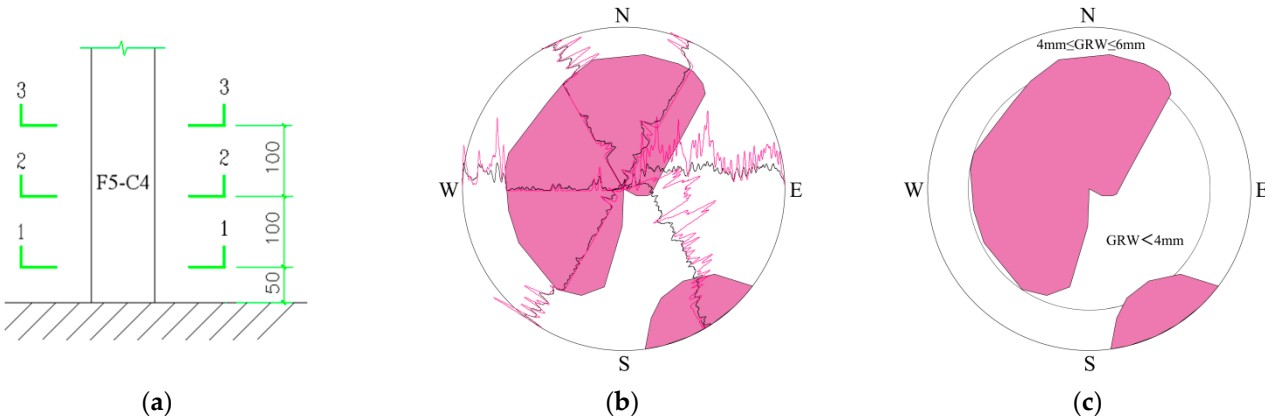

**Figure 23.** Micro-drilling test of F5-C4; (**a**) testing sections; (**b**) decay area determined by micro-drilling maps; (**c**) different GRW areas.

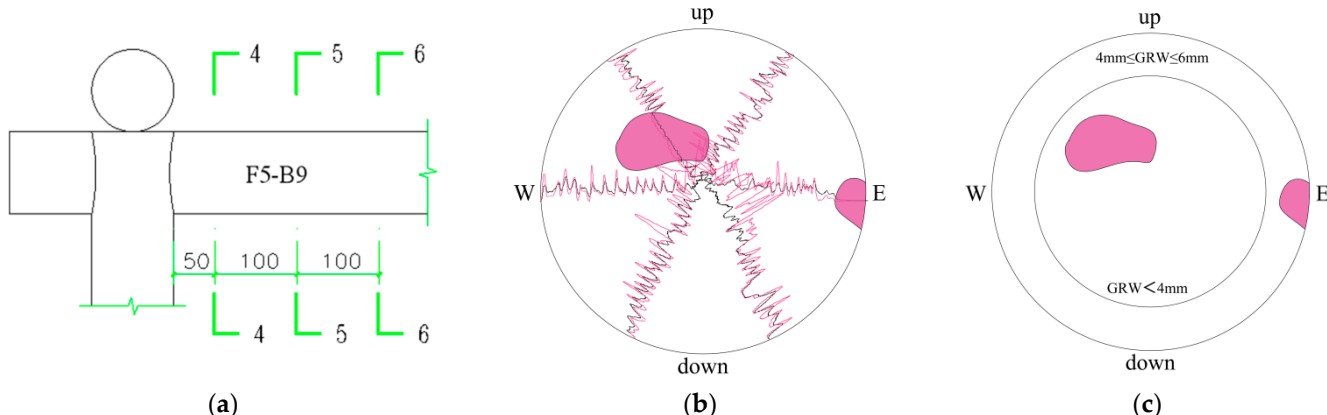

**Figure 24.** Micro-drilling test of F5-B9; (**a**) testing sections; (**b**) decay area determined by micro-drilling maps; (**c**) different GRW areas.

In the frame F5, the bases of some columns (F5-C1 to F5-C5) appear to have decay, and the ends of some beams (F5-B1, F5-B5 to F5-B7, F5-B9 to F5-B10) also have damage. While the rest of the components are in good condition. The decay grades of components in F5 were determined according to GB/T13942.2-1992 [1] and testing data (Figure 16b). Comparing the moisture content schematic diagram of the F5, the moisture content of the place where the decay occurs is higher, which further proves the correlation between the two. However, when the moisture content is more than 30%, the component reaches the so-called wet protective state. Comparing F1, F6, and F11, it can be found that although the moisture content of components in these frames is high, the decay only occurs in a few parts. Therefore, it cannot be believed that the parts with high moisture content of the components will decay. At the same time, the moisture content of the components fluctuate with the season, weather, and other objective factors, which causes a certain degree of difference in detection. The detection time was summer, the Yangzhou area is hot and rainy in summer, and the air humidity is high. Therefore, the moisture content of the component tested during this time must be higher than its annual average value.

For the overall structure, more than half of the columns on the ground floor have visible decay, but the degree of decay is not high. The beams' condition is basically intact, and there is no obvious surface decay. Most of the components in the structure are in decay grades I and II, only a few of the components around the courtyard or buried in the walls belong to grades III and IV (Figure 25).

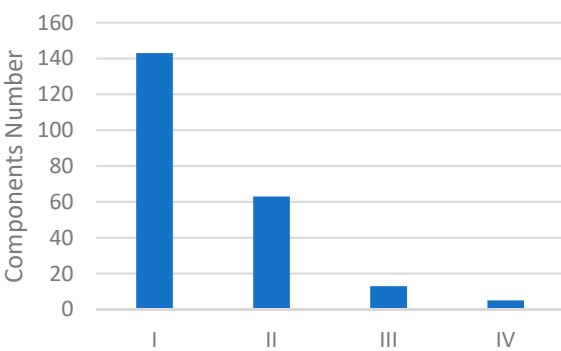

**Figure 25.** The number of components in different decay grade.

3.2.4. Estimating Material Properties Parameters

The estimation of the material property parameters of timber components is also illustrated with the column F5-C4 and beam F5-B9 as an example.

The diameter of F5-C4 is about 180 mm, and the overall condition of the component is good. According to the micro-drilling resistance map, the effective section area (S = 15209 mm$^2$) can be determined by connecting the decayed area in a specific way. Corresponding to the resistance map, the area ranges of different GRW are divided (as shown in Figure 23c: $S_1$ = 6906 mm$^2$, $S_2$ = 8303 mm$^2$, $S_3$ = 0 mm$^2$, $S_4$ = 0 mm$^2$). According to Formula (1) and (2), the mechanical parameters of components can be calculated: $MOE_{F5\text{-}C4}$ = 10,435.25 MPa, $MOR_{F5\text{-}C4}$ = 47.05 MPa.

The diameter of F5-B9 is about 180 mm, and the surface of the beam is soft and without obvious decay. Its effective section area is S = 23,873 mm$^2$, and $S_1$ = 12,215 mm$^2$, $S_2$ = 11,658 mm$^2$, $S_3$ = 0 mm$^2$, $S_4$ = 0 mm$^2$ (Figure 24c). According to Formula (1) and (2), the mechanical parameters of components can be calculated: $MOE_{F5\text{-}B9}$ = 10,460.02 MPa, $MOR_{F5\text{-}B9}$ = 47.41 MPa.

According to the unified calculation and analysis, the *MOE* of the whole structure is between 10,210.14 MPa and 10,507.61 MPa, and the *MOR* of it is between 42.47 MPa and 47.96 MPa. According to the estimated material parameters, the main timber structure components of the Fujiu Zhou house are basically in a safe condition, and there is no risk of damage or collapse for the time being.

## 4. Discussion and Conclusions

By means of on-site investigation, measurement and laboratory analysis, many basic data obtained from wood species identification, appearance survey, moisture content tests and micro-drilling resistance tests can be used to check and distinguish the timber species, the types of defects, and the degree of decay. The results can provide scientific support for further scientific evaluation.

Referring to the existing comprehensive evaluation criteria and methods, the various factors affecting the working performance of timber components and their correlation are analyzed, which can effectively classify the structural properties of each component of historical timber building scientifically.

Based on the correlation between the GRW of timber and related physical and mechanical properties, combining with the classification grade of timber component structure, the regions of different GRW area and decay position within the components can be divided. Then the *MOE* and *MOR* of the component can be calculated according to the area weights of different regions.

For the case study of the Fujiu Zhou house, the species of its main structure is called *Cunninghamia lanceolata* (Chinese fir). The decay area of the structure has a high relationship with moisture content. With the micro-drilling resistance test, the decay grade of most components in the structure belongs to grade I and II. The *MOE* of the whole structure is between 10,210.14 MPa and 10,507.61 MPa, and the *MOR* of it is between 42.47 MPa and 47.96 MPa. The main structure is basically stable.

In general, this study proposes a series of scientific methods that can be used to assess the condition of timber components in historical buildings. In this method, on-site investigation, laboratory experiments, and even physical and mechanical parameter estimation methods can be mastered by all kinds of non-professionals who are engaged in historical building protection research. It can also increase the participation of people in the protection of historical timber structures, no longer shut out non-professionals, and help and attract more people with lofty ideals to participate in the protection of architectural heritage.

It is worth noting that under the premise of fully complying with the requirements of the architectural heritage protection work, it is impossible to truly realize the specific test and detection of the mechanical parameters of the timber components. The parameter estimation method proposed in this study is a relatively rough method. The purpose and advantages of this method are simple, fast, convenient, and learnable. It is expected to play an auxiliary role in research for various professional and non-professionals. At the same time, due to the limitation of basic research results and field test conditions, it is necessary to make supplementary analysis according to specific conditions to apply this method to other specific projects.

**Author Contributions:** Conceptualization, data curation, formal analysis, investigation, methodology, resources, visualization, writing-original draft, writing-review and editing, P.W.; methodology, resources, writing-review and editing, S.L., N.M. and S.P.; writing-review and editing, G.M. All authors have read and agreed to the published version of the manuscript.

**Funding:** This research was funded by National Natural Science Foundation of China, grant number 51478409, 51338001; International cooperation research program of Yangzhou, grant number YZ2019148; Program of 100 Foreign Experts in Jiangsu Province, grant number JSB2017029.

**Acknowledgments:** The author Peixuan Wang who is studying in Politecnico di Milano was funded by the China Scholarship Council. Thanks to S. Lazzeri for the contribution to this paper.

**Conflicts of Interest:** The authors declare no conflict of interest.

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
