# Peer review of "Comprehensive Evaluation Method of Historical Timber Structural Building Taking Fujiu Zhou House as an Example"

_forests, doi:10.3390/f12091172_

Round 1

Reviewer 1 Report

General comments:

As a general comment, I found the paper interesting although in my opinion the paper needs to extend the discussion and state clearly why the findings are important. Discussion and conclusion relate to well-known statements.

The title of an article should be changed. In reviewer’s opinion it should be extended by indicating "on selected example". In the opinion of the reviewer, the article is not a scientific article but rather a case study.

Line 19 – in the first place where the name of the building appears, it is advisable to specify exactly where Fujiu Zhou house is located

Line 19 - please indicate how the grade of timber components was determined, what kind of timber components were analysed?

Line 50 -  please improve the quality of photo 2, it is insufficient even in colour printout

Line 51 - a typical approach in the conservation analysis is from the overall system to the detail. In the opinion of the reviewer, a review of the current state of knowledge should be an extension of that thought

Line 84 – What are the timber species most used in Italy?  Why is the methodology not general and applicable to all types of wood?

Lines 96 – 99 - there is no novelty in presented aim of paper. It should be more precisely indicated what other / new aspects are undertaken compared to those analyzed by other researchers. In the opinion of the reviewer, there is no novelty in the work at all. The work is strictly a case study, not a scientific article.

Line 136 – 152 - lack of discussion in relation to very limited and local range of micro-drilling technique, need to be complemented

Line 154 – I disagree with sentence in lines 154-155. Species identification is not crucial for physical and mechanical properties determination.

Line 186 - what is the significance of the Canadian standard for sorting Italian or Chinese lumber?

Line 188 – “Relevant research results…” - Canadian standard has been shown as a reference. Typo? additionally, it would be appropriate to indicate SEVERAL publications, since the topic has already been discussed, not only one reference.

Line 194 – 205 - Methodical error. The values of MOE and MOR are highly dependent on the cross-section of the element being analyzed (scale effect). It was not indicated what cross-section was the subject of the research in [21]. Authors cannot directly relate the obtained values in [21]  to their applications without correction. Authors wrote that the table shows “relationship values between GRW, MOE and MOR”, but only R or R2 coefficient is the factor responsible for the correlation between the values.

Additionally, what about the statistics ? COV is not enough and strictly indicates no significant differences between e.g. average MOE values for GRW <4 mm and 4mm <= GRW <= 6mm.

Line 203 – 204 - the sentence needs to be corrected

Line 206 – Please correct the unit – MPa

Lines 208 – 215 - double dashes -Typo ?

Lines 243 – 252 - Was a sample actually taken to determine the species of wood microscopically? why figure 7 is authored by S. Lazzeri ? What about a reference or authorship?

Line 283 - with which device was the MC determined? what was the ambient humidity? measurements were made in what period (winter / autumn / summer?)

Line 313 – True, but not necessary. This dependence is known, but above 30% MC we deal with the so-called wet protective state. Additionally - and importantly, differences in MC and its fluctuation over time are important.

Line 330 – 180 mm ?

Lines 341 – 366 - There is no novelty in the work. Convertion the GRW into MOR and MOE is questionable. R2 coefficient between GRW and MOR and MOE is at the level of about 0.2-0.3, which actually proves the lack of dependence between the features, especially in the case of samples with a relatively large cross-section, full-size elements in technical scale.

Author Response

Point 1: As a general comment, I found the paper interesting although in my opinion the paper needs to extend the discussion and state clearly why the findings are important. Discussion and conclusion relate to well-known statements. 

Response 1:

We thank the reviewer for his/her positive comment on the paper. More content describing why these findings are important has been added to the “Abstract” Section (lines 17-19, in red), “Introduction” Section (lines 118-126, in red), and “Discussion and conclusion” Section (lines 443-450, in red). This paper mainly proposes a scientific method for comprehensive evaluation of timber components that can be quickly mastered by all kinds of non-professionals. It aims to enhance the interdisciplinary nature of the research on architectural heritage protection and attract and help more non-professionals to devote themselves to scientific research.

Point 2: The title of an article should be changed. In reviewer’s opinion it should be extended by indicating "on selected example". In the opinion of the reviewer, the article is not a scientific article but rather a case study.

Response 2: 

We thank the reviewer for his/her kindly comment on the title, and we also are very grateful to reviewers for the definitions of scientific papers/case studies. This is a comprehensive question. This paper uses a lot of space to detail how to apply the method summarized in it to a timber structure case. At the same time, the paper also summarizes a series of comprehensive evaluation methods for the condition of timber components, especially for the estimation of physical and mechanical parameters. The author hopes to prove the scientific nature of the research method proposed in this paper through case study. According to the suggestion from reviewer, the title of the paper has been modified to "Comprehensive Evaluation Method of Historical Timber Structure Building Taking Fujiu Zhou House as an Example" (lines 2-3, in red).

Point 3: Line 19 — in the first place where the name of the building appears, it is advisable to specify exactly where Fujiu Zhou house is locate.

Response 3:

We thank the reviewer for his/her suggestion. The location of Fujiu Zhou house has been added in the abstract (lines 20-21, in red).

Point 4: Line 19 — please indicate how the grade of timber components was determined, what kind of timber components were analysed?

Response 4:

We thank the reviewer for his/her suggestion, and the suggestion helps us enrich the completeness of the abstract. The method which can determine the grade of timber components and the main body of analysis has been added in the paper (line 21-24, in red). 

Point 5: Line 50 — please improve the quality of photo 2, it is insufficient even in colour printout.

Response 5:

We thank the reviewer for his/her suggestion. We chose two more suitable photos for replacement, sharpened the photos, and improved the brightness and contrast of them (Figure 2, in red).

Point 6: Line 51 — a typical approach in the conservation analysis is from the overall system to the detail. In the opinion of the reviewer, a review of the current state of knowledge should be an extension of that thought.

Response 6:

We thank the reviewer for his/her suggestion. We have added relevant content and references in the paper (lines 55-61, lines 70-85, in red).

Added reference (lines 475-482, in red) :

Dong Q.Q., Zhao T.F., Ni H.R., Wu Z.W., Wang A.S. Discussion on damage identification method of historical building timber structure. Guangdong Building Materials 2021, 37(08), 71-73.

Li C., Zhao Q.S., Ni G.Q., Liu X.Y. The life expectancy of timber structure of historical building. Collection and investment 2021, 12(03), 78-80.

Ni Y., Tang G.L., Zhang Z.P., Wang C. Research on the protection and repair methods of historical timber structural Build-ings—taking Diaodong Pavilion in Jingxian as an example. Value Engineering 2019,38(23), 238-240.

Wang Q.G. A Comparative Study on the Protection Technology of Historic Buildings in China and German. Ph.D. thesis, Zhengzhou University, Zhengzhou, China, 2018.

Point 7: Line 84 — What are the timber species most used in Italy?  Why is the methodology not general and applicable to all types of wood?

Response 7:

We thank the reviewer for his/her questions. In general, the most use of timber species is very complicated, and it needs to fully consider factors such as building function, geographical environment, climate, biological species, and designer's wishes. The timber species commonly used in Italy include: fir, maple, cherry, camphor, etc., and modern buildings mostly use glulam.

As for the reviewer's question about this sentence “Clearly, the methodology is applicable to the timber species most used in Italy and to the local timber structure typologies” in the paper, we gave the following answer. Here is a literature review of existing research methods. This method is derived from the first document published in Italy to describe the survey methodology for historical timber structures. Therefore, the authors summarized this method for the document that is suitable for the most commonly used timber species and local timber structure in Italy. Taking into account the differences in the architectural background, cultural background, and geographical environment of the East and the West, the authors dare not presumptuously infer whether the method is universal in the literature review part.

Point 8: Lines 96-99 — there is no novelty in presented aim of paper. It should be more precisely indicated what other / new aspects are undertaken compared to those analyzed by other researchers. In the opinion of the reviewer, there is no novelty in the work at all. The work is strictly a case study, not a scientific article.

Response 8:

We thank the reviewer for his/her kindly comment, the comment prompted us to think more deeply about the innovative points and themes of the paper. The innovative points of our paper is to summarize a set of technical methods for estimating the physical and mechanical properties of timber components that can be quickly learned and used by various non-professionals. We has did the modify of this part in the paper (lines 17-19, lines 118-126, lines 443-450, in red).

Point 9: Line 136-152 — lack of discussion in relation to very limited and local range of micro-drilling technique, need to be complemented.

Response 9:

We thank the reviewer for his/her important suggestions. Taking into account the completeness and richness of the paper, we have made a more detail supplemental description of the detection method of component moisture content and the micro-drilling resistance.

The content about moisture content test has been added in the paper (lines 161-172, in red).  The content about micro-drilling technique has been added in the paper (lines 173-177, lines 195-210, in red).

Point 10: Line 154 — I disagree with sentence in lines 154-155. Species identification is not crucial for physical and mechanical properties determination.

Response 10:

We thank the reviewer for his/her important comment. We very much agree with the comments, and apologize for the imprecise use of words. We have modified this part of the content (lines 212-214, in red). 

Point 11: Line 186 — what is the significance of the Canadian standard for sorting Italian or Chinese lumber?

Response 11:

We thank the reviewer for his/her question.

Canada is currently an important timber producing country in the world. The development of timber construction in this country started earlier and developed rapidly, and has been at the forefront of the world. A variety of standards related to timber and timber structural buildings published in Canada are obtained through long-term research experience.

The work experience of most of the authors who participated in this article indicated that Canadian standards are an important reference in the research and protection of timber buildings in Italy. Combining the existing frontier references can promote the in-depth study of Italian timber architecture and play a positive role in the designation of its own national standards.

As far as China is concerned, Chinese timber structure buildings appeared earlier, but the research started relatively late, which is far behind western developed countries. Although in ancient China, there were certain specifications for the selection and use standards of timber structure buildings, but these requirements were mainly passed on by the craftsman population, and there was no standard document that could form a large number of specifications. After entering the modern era, due to the late start of development, China has revised a considerable part of the standard specifications for timber selection based on the research experience of existing scholars, but there is still room for improvement in its completeness. Therefore, we chose to refer to the advanced Canadian standards, hoping to be able to use a more mature method to classify and distinguish timber.

Point 12: Line 188 — “Relevant research results…” - Canadian standard has been shown as a reference. Typo? additionally, it would be appropriate to indicate SEVERAL publications, since the topic has already been discussed, not only one reference.

Response 12:

We thank the reviewer for his/her kindly suggestions. Yes, here is a labeling error, and we have corrected the content (line 253, in red). At the same time, according to the advice, we have also added relevant references.

Added reference (lines 523-531, in red):

Duan X.F., Wang P., Zhou G.W., Gao C.Y. Preliminary study on the stress wave technology to detect the residual elastic modulus of historical building timber components. Journal of Northwest Forestry University 2007, (01), 112-114.

Zhu L. Research on testing technology of mechanical properties of historical building timber components based on stress wave. Ph.D. thesis, Beijing Forestry University, Qinghua east road, Haidian district, Beijing, China, 2012.

Zhang H.J., Zhu L., Sun Y.L., Wang X.P. Research on testing methods of main mechanical properties of historical building timber component materials. Journal of Beijing Forestry University 2013, 33(05), 126-129.

Zhu L., Zhang H.J., Sun Y.L., Wang X.P., Yang H.C. Non-destructive testing of mechanical properties of Korean pine timber components based on stress wave and micro-drill resistance. Journal of Nanjing Forestry University (Natural Science Edition) 2013, 37(02), 156-158.

Point 13: Line 194-205 — Methodical error. The values of MOE and MOR are highly dependent on the cross-section of the element being analyzed (scale effect). It was not indicated what cross-section was the subject of the research in [21]. Authors cannot directly relate the obtained values in [21]  to their applications without correction. Authors wrote that the table shows “relationship values between GRW, MOE and MOR”, but only R or R2 coefficient is the factor responsible for the correlation between the values.

Response 13:

We thank the reviewer for his/her careful review and the important comments. We are very sorry for the improprieties in the description and expression here. In the initial description, the description of this part of the content was absolute, and the author has modified this (lines 243-258, in red)

  • For“The values of MOE and MOR are highly dependent on the cross-section of the element being analyzed (scale effect)”:

First of all, in terms of mechanical parameters, materials such as timber and concrete are very different. The differences between individual materials are very large. Unless the materials are measured individually by precision instruments and confirmed by later experiments, we cannot assert the overall physical and mechanical properties of a certain batch of timber. Considering this factor, in the research and protection of historical structures, the research on the mechanical properties of components is even more difficult. In accordance with the requirements of historical architectural heritage protection, in the research process, we strive not to cause major damage and impact on the building structure. Therefore, we cannot test and analyze a single component, let alone infer the mechanical properties of the overall frame through the compression and tensile tests of an existing building. In this case, the authors just hope to judge the internal structure of the timber through the micro-drilling resistance instrument map, and then establish the correlation between the cross section of the component and the mechanical parameters (lines 253-256, in red ).

  • For “the cross-section of reference 21(now is reference 25)” and “the issues related to GRW, MOE, and MOR”:

The estimation of the mechanical parameters of timber components is one of the links in the evaluation of the overall performance of timber structures, aiming to enable more non-structural professionals to quickly and easily grasp this estimation method. Regarding the cross-section of reference 21 (now is reference 25) proposed by the reviewer and the issues related to GRW, MOE, and MOR, we acknowledge the differences that may arise from the use of the methods in the text.

Therefore, according to the reviewer’s opinion, all the terms used in the “extraction” of mechanical parameters in the article have been changed to "estimate". And in the conclusion of the paper, some of the shortcomings of this method are reiterated, indicating the need for local conditions to be improved according to specific conditions. At the same time, the particularity of timber materials is emphasized again. Unless the research components are individually tensile and compressive tests, the method proposed in this paper can only be a speculative value, which plays a supplementary role in building repair and protection (lines 451-457, in red).

Point 14: Additionally, what about the statistics ? COV is not enough and strictly indicates no significant differences between e.g. average MOE values for GRW <4 mm and 4mm <= GRW <= 6mm.

Response 14:

We thank the reviewer for his/her kindly suggestions. We agree with the reviewer’s thinking. As mentioned in the response 13, this paper aims to propose a comprehensive estimation method that is easy for non-professionals to master. Considering this part, we ignores the difference.

Point 15: Line 203-204 — the sentence needs to be corrected.

Response 15:

We thank the reviewer for his/her careful review. We apologize for our negligence, and part of this sentence is repeated. We had deleted it and corrected the sentence (lines 277-279, in red). 

Point 16: Line 206 — Please correct the unit - MPa.

Response 16:

We thank the reviewer for his/her careful review. We had corrected the unit MPa (line 280, Table 3, in red).

Point 17: Lines 208- 215 — double dashes - Typo?

Response 17:

We thank the reviewer for his/her careful review. We had corrected the dash (lines 282-288, in red).

Point 18: Lines 243-252 — Was a sample actually taken to determine the species of wood microscopically? why figure 7 is authored by S. Lazzeri? What about a reference or authorship?

Response 18:

We thank the reviewer for his/her important comment. Yes, the results are from on-site test and laboratory tests. The figure 7 was done by S. Lazzeri who is a colleague of the two co-authors of this article. S. Lazzeri was only engaged in the only work in this part, and had no other contributions to the project research methods, on-site testing, and article writing. So we did not includ S. Lazzeri as the author after discussion, and we decided to put S. Lazzeri in the “Acknowledgments” Section (line 469, in red).

Point 19: Line 283 — with which device was the MC determined? what was the ambient humidity? measurements were made in what period (winter / autumn / summer)?

Response 19:

We thank the reviewer for his/her question. The model of the wood moisture meter is JK W10. Honestly, we did not use the instrument to detect the ambient humidity, but recorded the relative humidity by querying the local weather forecast. The detection time: from 18-08-2018 to 19-08-2018, summer, the relative humidity of these two days are all 76% . We also organized this part of the content and added it to the paper (lines 335-337, in red).

Point 20: Line 313 — True, but not necessary. This dependence is known, but above 30% MC we deal with the so-called wet protective state. Additionally - and importantly, differences in MC and its fluctuation over time are important.

Response 20:

We thank the reviewer for his/her useful suggestions. Yes, we did not mention the impact of the wet protective state and the influence of detection time in the analysis of the case study results. This is our negligence. This part of the content has now been added to the paper (lines 387-396, in red).

Point 21: Line 330 — 180 mm?

Response 21:

We thank the reviewer for his/her careful review. We had corrected the unit (line 412, in red).

Point 22: Lines 341-366 — There is no novelty in the work. Convertion the GRW into MOR and MOE is questionable. R2 coefficient between GRW and MOR and MOE is at the level of about 0.2-0.3, which actually proves the lack of dependence between the features, especially in the case of samples with a relatively large cross-section, full-size elements in technical scale.

Response 22:

We thank the reviewer for his/her comments. The innovative points of our paper is to summarize a set of technical methods for estimating the physical and mechanical properties of timber components that can be quickly learned and used by various non-professionals. And we had added this part in the paper (lines 17-19, lines 118-126, lines 443-450, in red).

We agree with the reviewer's view on the deficiencies related to the estimation of the mechanical parameters of timber components. We thank the reviewers for their consideration of the special circumstances of large cross-section, full-size elements components, which can help us enrich the completeness and rigor of the paper. As mentioned in response 13, what we propose here is an estimation method. The estimation of the mechanical parameters of timber components is one of the links in the evaluation of the overall performance of timber structures, aiming to enable more non-structural professionals to quickly and easily grasp this estimation method. In the case of insufficient dependence between features, we revised the content of the paper and pointed out this difference in the method section (lines 244-258, in red). According to the reviewer’s opinion, all the terms used in the “extraction” of mechanical parameters in the article have been changed to "estimate". At the conclusion, we also reiterated the possible research differences and emphasized that the use of this method needs to be adapted to local conditions, and appropriate adjustments should be made according to the actual situation (lines 451-457, in red).

PS:

There have been minor revisions in other parts of the paper, all of which are shown in red.

We would like to thank the referee again for taking the time to review our manuscript. Your valuable suggestions have greatly improved the scientific nature of our paper and pointed out a more distinct research direction for us.

We are also very grateful to the editor for the valuable time and energy on the paper.

Reviewer 2 Report

Methods of assessing the condition of the construction of wooden buildings are not among the frequently published topics. This is a very demanding issue not only in terms of conducting the expertise itself but also in terms of the possibility of presenting the results in the form of a scientific article. For the purposes of the article, the authors have chosen a specific application of proven scientific methods for evaluating the condition of wooden structures, taking into account their local specifics - a specific wooden structure. Therefore, I understand the meaning of the article more from a methodological point of view as a good example of the application of these methods.

The article from the point of view of structure is processed according to the rules of a scientific article. The theoretical analysis is sufficient as well as the scope of the cited literature (further addition would be only an artificial increase). The methodology is elaborated in sufficient detail and is based on proven methodological procedures of other authors. The results are processed adequately to the topic. If the reader studies the methodology carefully enough, he has no problem understanding the pictures. The textual evaluation corresponds to the displayed results. Conclusion and discussion is sufficient.

Author Response

Point 1: Methods of assessing the condition of the construction of wooden buildings are not among the frequently published topics. This is a very demanding issue not only in terms of conducting the expertise itself but also in terms of the possibility of presenting the results in the form of a scientific article. For the purposes of the article, the authors have chosen a specific application of proven scientific methods for evaluating the condition of wooden structures, taking into account their local specifics - a specific wooden structure. Therefore, I understand the meaning of the article more from a methodological point of view as a good example of the application of these methods.

Response 1:

We are very grateful to the reviewer for his/her affirmation of this article, and at the same time express our heartfelt thanks to the reviewers for his/her time and energy during the review process. As the reviewer believes, the research method for estimating the material parameters of timber components in this paper is an innovation, and an attempt is made to demonstrate the feasibility of the research method through a case.

Point 2: The article from the point of view of structure is processed according to the rules of a scientific article. The theoretical analysis is sufficient as well as the scope of the cited literature (further addition would be only an artificial increase). The methodology is elaborated in sufficient detail and is based on proven methodological procedures of other authors. The results are processed adequately to the topic. If the reader studies the methodology carefully enough, he has no problem understanding the pictures. The textual evaluation corresponds to the displayed results. Conclusion and discussion is sufficient.

Response 2: 

We are very grateful to the reviewer for trying to interpret the paper from both professional and non-professional perspectives, fully considering the readability of this article for non-professional researchers. This paper added a considerable part of references during the major revisions, and described the research methods in more detail (in red in the new manuscript). It is hoped that through the increase of this part of the content, the scientificity and readability of this paper will be enhanced, so that the research can serve all kinds of professionals and non-professionals.

PS:

All the revised parts of the paper are marked in red

We would like to thank the referee again for taking the time to review our manuscript. Your valuable suggestions have greatly improved the scientific nature of our paper and pointed out a more distinct research direction for us.

We are also very grateful to the editor for the valuable time and energy on the paper.

Round 2

Reviewer 1 Report

The authors have replied to all comments and remarks. I feel satisfied with the authors' answers and corrections. In my opinion, the article gained in value. However, I wish that non-professionals did not independently deal with matters related to the conservation of historic buildings.